# Predicting the intention to adopt wearable payment devices in China: The use of hybrid SEM-Neural network approach

Li Luyao[1], Abdullah Al Mamun ®[2]*, Naeem Hayat[3], Qing Yang ®[2], Mohammad Enamul Hoque[4], Noor Raihani Zainol ®[5]

**1** UCSI Graduate Business School, UCSI University, Kuala Lumpur, Malaysia, **2** UKM - Graduate School of Business, Universiti Kebangsaan Malaysia, Bangi, Selangor, Malaysia, **3** Global Entrepreneurship Research and Innovation Centre, Universiti Malaysia Kelantan, Kota Bharu, Malaysia, **4** BRAC Business School, BRAC University, Dhaka, Bangladesh, **5** Faculty of Entrepreneurship and Business, Universiti Malaysia Kelantan, Kota Bharu, Malaysia

\* almamun@ukm.edu.my, mamun7793@gmail.com

**Data Availability Statement:** All relevant data are within the paper and its Supporting information files.

## Abstract

Wearable payment devices (WPD) are gaining acceptance fast and transforming everyday life and commercial operations in China. Limited research works were conducted on customers' adoption intentions to obtain a real image of the evolution of WPD in China. This study aims to investigate the effects of Performance Expectancy (PE), Effort Expectancy (EE), Social Influence (SI), Facilitating Conditions (FC), Hedonic Motivation (HM), Perceived Trust (PT), and Lifestyle Compatibility (LC) on the intention to adopt WPD among Chinese consumers by expanding unified theory of acceptance and use of technology with two impelling determinants (i.e. PT and LC). Using an online survey, empirical data were collected from 298 respondents in China. In a two-stage data analysis, partial least squares structural equation modelling (PLS-SEM) were employed to analyse the causal effects and associations between independent and dependent variables, whereas artificial neural networks (ANN) were used to evaluate the research model prediction capability. The (PLS-SEM) findings indicated that PE, SI, FC, HM, LC, and PT had substantial positive impacts on adoption intention, whilst EE had no impact on adoption intention among Chinese consumers. The ANN analysis proved the high prediction accuracy of data fitness, with ANN findings highlighting the importance of PT, FC, and PE on the intention to adopt WPD. It was suggested that the study findings assist WPD service providers and the smart wearable device industry practitioners in developing innovative products and implementing efficient marketing strategies to attract the existing and potential WPD users in China.

## Introduction

A particular emphasis is placed on the advancement of mobile networks, mobile device integrations, big data analytics, and Internet of Things (IoT) technologies in the business, commerce, financial, corporate, and socio-economic environment across a wide variety of strategic

**Funding:** The author(s) received no specific funding for this work.

**Competing interests:** The authors have declared that no competing interests exist.

IT development lines. Given the continuous improvement in mobile payment (m-payment) technology, wearable payment technology is available to play its role in the marketplace. Mobile technologies, such as smartphones, IoT, and smart wearable devices (e.g., smart-watches, wristbands, and rings), have transformed conventional payment systems from card and cash-based to contactless payment systems [1]. Currently, payment technology allows extending payment capabilities beyond mobile phones to a broader and more diversified system of internet-connected wearable payment devices (WPD) through the rapid growth of the IoT [2].

Wearable payment is a method of contactless payment that involves using wearable devices enabled through the Near Field Communication (NFC) protocol [3]. Wearable devices facilitate paying for products and services at any time and place using smart wearable devices (e.g., smart-watches, rings, fitness-trackers, wrist-bands) that are physically attached to the users [4]. The most recent technologies for WPD include Apple Watch and digital rings, which are projected to eventually replace m-payments and card payments [5]. Another form of WPD is the smart stamp combined with an NFC-enabled smartphone, allowing users to make card-less NFC payments [6]. Notably, wearable payment devices are gaining higher popularity as a result of sophisticated technology, which contributes to the fast, convenient, and secure payment method in several settings [4]. Wearable devices have been established worldwide as one of the most important IoT products, with sales of over 141 million units in 2019 [7]. According to the surveys on contactless payment systems conducted in 2019, 67% of customers were using m-payments, with a 19% increase in customers occurring in Thailand, 24% increase in Vietnam, and 20% increase in Middle East nations [8]. The number of Chinese smartphone users who employed M-payments was comparably higher than in many other countries, accounting for 81% of M-payment user worldwide [9]. According to the 47[th] statistics report on the growth of the internet in China, the total number of m-payment users in China should have reached 853 million by December 2020, which was 86.5% of the country's total mobile internet users [10]. Following the analysis of the statistics of the growth in the use of smart wearable devices and the use of contactless payments in China, a positive prospect for WPD may be anticipated among Chinese users. A wearable payment system remains a relatively new idea in China, which requires further investigation to analyse its behavioural intention and actual adoption among users.

Despite the usefulness and benefits of adopting wearable payment devices, there is a lack of study on wearable payment device adoption behaviour considering they are considerably in the early stages of commercialization. Interestingly, research in the context of China on m-payment usage [11–13] and wearable technology adoption [14] has sought to uncover the factors driving technological product and service adoption. Furthermore, while m-payment and WPD serve the same purposes, there are significant technological and procedural differences that distinguish WPD from m-payment [2]. Additionally, an individual's behavioural factors toward different products and services are totally subjective, and thus their adoption behaviour is likely to be different for each. So, understanding of m-payment of adoption behaviour, it may not be useful for product innovators and shoppers of WPD. Identifying research gaps in wearable payment device adoption behaviour, the current study investigates factors impacting WPD adoption in China.

To explore the factors of the intention to use contactless payments, previous studies focused on behavioural intention models and theories such as the Technology Acceptance Model (TAM) [15], Diffusion of Innovation Theory (DOI) [16], the Unified Theory of Acceptance and Use of Technology (UTAUT) [17–19], and UTAUT2 [20]. However, the impacts of lifestyle compatibility, perceived risk, and perceived trust on the adoption of new technology

were overlooked in the UTAUT and TAM [21]. Hence, lifestyle compatibility and perceived trust were determined as the important, influential factors in this study.

The study employed the UTAUT2 components to assess the theory's applicability to the WPD in the Chinese context, given that it is a widely adopted theory to build an understanding of the external and internal drivers of any new technology adoption intention [21]. Moreover, this study applied a relatively new research methodology, which consisted of two phases. A structural equation model (SEM) was applied in the first phase to assess the substantial effects of factors on WPD adoption. Following that was the second phase, which assessed the importance of the determinants using the artificial neural network (ANN) model.

## Literature review

### Theoretical foundation

The intention to use any new technology is a prerequisite for adopting it. Several earlier studies focused on three primary types of determinants, namely technological factors, human factors, and environmental variables. Most of these determinants were drawn from the "unified theory of acceptance and use of technology (UTAUT/UTAUT2)" and "Technology Acceptance Model (TAM)" [15, 19, 22–24]. Despite the tremendous recognition of UTAUT [25], Venkatesh et al. [26] expanded the UTAUT model with the addition of three new factors, namely "price value", "hedonic motivation", and "habit", resulting in the establishment of the UTAUT2 theory. Previous studies revealed that compared to the original UTAUT, the extensions offered in the UTAUT2 resulted in a substantial increase in the explained variance (e.g., 56% to 74%) of the intention to adopt new technologies [27]. Taking all these facts into account, the UTAUT2 model was used in this study to examine the factors influencing the consumers' intentions to embrace WPD in the context of China.

The majority of earlier research studies [20, 23, 24, 28] found that price value was not a significant predictor of adoption intention in the context of an upgraded technology with contactless payments. Following the aforementioned literature results and judgments, price value was not considered in this study. Customers should acquire high experience in adopting technology to investigate the function of habit [29]. Furthermore, a relatively new technology that has not acquired significant market acceptance may be unable to establish a habit [28, 30]. Notably, many of the participants in this study were potential users who had yet to use WPD or had made attempts to use it. As a result, the researcher decided not to include the price value and habit in the current study model.

UTAUT and UTAUT2 were expanded by several studies [18, 19, 27] through the incorporation of perceived trust as an influencing factor of the intention to adopt contactless payments. According to other earlier studies, lifestyle compatibility was among the most important indicators of the intention to adopt contactless payment technologies [18, 19, 23, 31]. Therefore, this study evaluated the impacts of perceived trust and lifestyle compatibility on the intention to adopt WPD and UTAUT2 model components to explore the intention to adopt WPD among Chinese consumers. The research model is offered in Fig 1.

### Hypothesis development

**Performance expectancy (PE).**   Performance expectancy refers to the degree to which the use of new technology could assist consumers in performing certain operations [26]. In the case of contactless payments, PE indicates customers' judgments of improved performance as a result of employing payment services, such as payment expediency and accuracy, fast response, and service efficiency [27]. To make a payment using WPD, the user simply needs to install payment software on their smart wearable gadgets and wave it over the payment

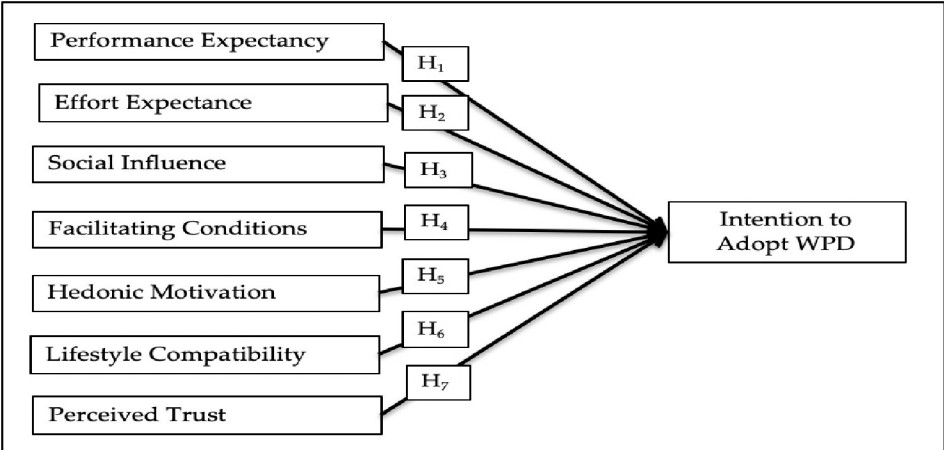

**Fig 1. Research framework.**

terminals [2]. This method could reduce the time required to complete a traditional transaction. Furthermore, PE was found as a crucial antecedent to different contactless payment-related research works, such as mobile banking [29], NFC based m-payment [15, 20], m-payments [24], e-wallet [8], and wearable payment [2]. The following hypothesis was developed in this study:

H1: PE positively influences the intention to adopt WPD.

**Effort expectancy (EE).**   Effort expectancy denotes the extent to which any new technology appears simple and uncomplicated to use due to its relation to customer adaptation [26]. Consumers would be interested in acquiring the desired outcomes when the new technologies are perceived to be easier to use and not burdensome [32]. Currently, consumers can make payments more easily, quickly, and safely due to the technological advancements in contactless payment [18]. Furthermore, EE could be improved through widely accessible M-payment service providers and the availability of the application on smartphones and other portable devices with device-independent apps [27]. Contactless payment systems that operate on wearable smart devices on the wrist by simply waving contribute to a convenient payment method, which allows customers to pay for purchases using single gestures [2]. Numerous empirical studies have proven that EE is a strong predictor of consumers' intent to use contactless payments, including M-payments [17, 24, 33], e-wallets [18], and wearable payment [34]. Therefore, to examine the impact of EE on the adoption intention of WPD, the following hypothesis was established:

H2: EE positively influences the intention to adopt WPD.

**Social influence (SI).**   Upon determining whether the adoption of new technology is required or vice versa, consumers emphasise the opinions of their close relatives who are identified as a social influence [25]. To illustrate, the information and motivation shown by other people around a consumer may hold a crucial role in enhancing their awareness of products and increasing their willingness to adopt the technology [29]. Social influence represents people's attitudes towards other people's expectations of the adoption of technology and their

motivation to meet those expectations [27]. Oliveira et al. [28] demonstrated the high importance of SI by stating that the adoption intention of m-payment is strongly affected by the suggestions and opinions of prominent people in society. Moreover, SI is a significant factor of behavioural intention in numerous contactless payment studies, which involved NFC-based M-payment [15, 20], e-wallet [18], and mobile money [27]. Following is the hypothesis proposed in this study based on an analysis of all prior studies on contactless payment technologies:

H3: SI positively influences the intention to adopt WPD.

**Facilitating conditions (FC).** Facilitating Conditions refer to a consumer's expectations about the availability of resources and support to complete a task using new technology [26]. It signifies the presence and availability of payment service resources (e.g., functional mobile network connection, transfer agents, operational skills, and financial resources) that allow the adoption of contactless payment technology [27]. M-payments require users to possess several essential abilities, such as operating a smartphone and running applications through it, sending or receiving reports, and connecting to a bank account [35]. Facilitating Conditions offers favourable and better resource sufficiency, a higher degree of service support, and lower financial cost, which leads to a substantial impact on contactless payments such as m-payment [21, 23, 24, 30], mobile-banking [29], and m-wallet [22]. Based on prior research findings and theoretical orientation, the following hypothesis was proposed in this study:

H6: FC positively influences the intention to adopt WPD.

**Hedonic motivation (HM).** Hedonic motivation denotes the intrinsic utilities (e.g., joy, playfulness, fun, and enjoyment) that are incorporated alongside extrinsic utilities in the UTAUT model [26]. The HM is the concept of perceived pleasure and satisfaction, which impacts the client's intention to use innovative technologies [36]. Furthermore, it has been found to possess a major role in assessing the level of adoption intention of new technology [26]. Similarly, the enjoyment or pleasure gained from utilising new technology promotes consumers' behavioural intention [29]. The HM was recorded as a significant predictor of adoption intention in several studies of contactless payment systems, including mobile banking [29], M-payment [20, 23, 30], and mobile-money [27]. Based on the previous research findings, the following hypothesis was established in this study:

H5: HM positively influences the intention to adopt WPD.

**Lifestyle compatibility (LC).** Consumers in societies acquire services and evaluate lifestyle norms and the values associated with the services [37]. Lifestyle Compatibility denotes the natural alignment of lifestyle choices, which creates a social image among peers [22]. Furthermore, it was also a strong predictor of an individual's willingness to accept new technology [31]. According to Oliveira et al. [28], customers are more likely to use m-payment technologies, provided the technologies are suitable for their lifestyle and social environment. Previous empirical research demonstrated that LC is a strong predictor of consumers' wish to use contactless payment systems such as M-payments [12, 23], e-wallets [18], and m-wallet [22]. The following hypothesis was developed in this study based on the previous research findings on contactless payment systems:

H6: LC positively influences the intention to adopt WPD.

**Perceived trust (TR).**   Individuals' wish, security, and confidence in a system to consistently meet their expectations and does not disappoint them is known as trust [38]. Customers' perceived trust and certainty in the mobile transaction service mandate that it must be secured from hackers, while a system free from fraud would assure an increased adoption intention [27]. Notably, customers' trust is vital in encouraging the adoption of contactless payment and forming a perception of the technology as more innovative and unique [29]. Individuals' trust is essential as it diminishes consumers' concerns, anxieties, and doubts, which improves adoption intentions [38]. Previous research on m-payments [1, 15, 33], e-wallets [18], mobile money [27], m-wallets [22], mobile-banking [29], and wearable payment [34] demonstrated the significant impact of PT. The following hypothesis was established in this study based on the evidence of contactless payment technologies from previous studies:

H5: TR positively influences the intention to adopt WPD.

## Research methodology

### Population and sample

This research employed a cross-sectional research design to observe the adoption intention of WPD in the context of China. Based on the studies incorporated new technological developments in contactless payments, the younger generation was deemed the ideal population to obtain an accurate image [39], given that youths are the advanced level adopters of new technologies, the internet, and mobile devices [40]. According to the International Labor Organisation [41], individuals aged 18 to 40 are considered a youth and working adults in the formal sector. Therefore, this study focused on youths in China who are 20 to 40 years old, are financially competent and have the decision-making ability to become users of WPD. In this quantitative study, a convenience sampling technique was applied to select the respondents as the sampling method was found to be economical. Thus, respondents could be addressed from any easily reachable part of the population. The minimal sample size in this study was calculated using G-Power Version 3.1 (with an effect size of 0.15, a power of 0.95, and eight predictors), while the sampling size amounted to 74 units. Data were obtained from 298 respondents through an online survey to avoid the drawbacks of a small sample size. Participants were assured that all information was kept confidential and their participation was voluntary. An online survey was conducted by posting the survey form at WJX site (http://www.wjx.cn/) from May 2021 to June 2021.

The local ethics committee (Universiti Malaysia Kelantan, Malaysia) ruled that no formal ethics approval was required in this particular case because this research did not collect any medical information, there was no known risk involved, did not intend to publish anyone's personal information, and did not collect data from underaged respondents. This study has been performed in accordance with the Declaration of Helsinki. Informed consent for participation was obtained from respondents who participated in the survey. The respondents who participated in the survey online (using google form) were asked to read the ethical statement posted on the top of the form (*There is no compensation for responding nor is there any known risk. In order to ensure that all information will remain confidential, please do not include your name. Participation is strictly voluntary, and you may refuse to participate at any time*) and proceed only if they agree. No data was collected from anyone under 18 years old.

### Measures of constructs

The survey questionnaire was adapted from previously tested and validated surveys, with minor changes made to fulfill this research's purposes. During the questionnaire development,

**Table 1. Measurements.**

| Factors | No of Items | Minimum | Maximum | Mean | Std. Deviation |
|---|---|---|---|---|---|
| PE | 6 | 1.330 | 5.000 | 3.968 | 0.724 |
| EE | 5 | 1.000 | 5.000 | 3.966 | 0.706 |
| SI | 5 | 1.000 | 5.000 | 3.370 | 1.093 |
| FC | 5 | 1.000 | 5.000 | 3.848 | 0.743 |
| HM | 5 | 1.000 | 5.000 | 3.885 | 0.761 |
| LC | 5 | 1.400 | 5.000 | 3.746 | 0.819 |
| PT | 5 | 1.000 | 5.000 | 3.573 | 0.778 |
| WPD | 6 | 1.000 | 5.000 | 3.805 | 0.812 |

**Note:** "PE: Performance Expectancy; EE: Effort Expectancy; SI: Social Influence; FC: Facilitating Conditions; HM: Hedonic Motivation; LC: Lifestyle Compatibility; PT: Perceived Trust; IWPD: Intention to Adopt WPDs.

**Source:** authors' data analysis.

clear, understandable, and unbiased wordings were used to ensure that the respondents were attracted to the questionnaire and concretely elaborated on their answers and thoughts. Six items were adopted from Gupta and Arora [24] and Rahman et al. [42] to measure performance expectancy. Four items under effort expectancy were derived from Gupta and Arora [24] and Talukder et al. [43]. To estimate the social influence, five items were adopted from Gupta and Arora [24], Rahman et al. [42], and Talukder et al. [43]. Furthermore, facilitating condition was evaluated with five items adopted from Gupta and Arora [24] and Rahman et al. [42], while five items were used to assess hedonic motivation, which was adopted from Gupta and Arora [24], Rahman et al. [42], and Talukder et al. [43]. Five items were derived from Talukder et al. [43] to measure lifestyle compatibility. Moreover, perceiver trust was obtained by adopting five items from Loh et al. [44] and Hussain et al. [23]. Four items were extracted from Gupta and Arora [24] and Rahman et al. [42] to measure the intention to adopt WPD. A five-point Likert scale was used for all items, which ranged from 'strongly disagree' to 'strongly agree'. All items adapted in this study are presented in S1 Appendix. Survey Instrument. The descriptive characteristics of all varieties are presented in Table 1 below.

## Common Method Variance (CMV)

In this study, the single factor accounted for 40.281% (below the recommended threshold of 50%), which approved the inconsequential influence of CMV [45]. CMV evaluated the current study by testing the full collinearity of all constructs [46]. All constructs regressed on the common variable. The recorded values of the variance inflation factor (as presented in Table 2) were less than 3.3, suggesting the absence of bias from the single-source data [46].

**Table 2. Full collinearity test.**

| | PE | EE | SI | FC | HM | LC | PT | IWPD |
|---|---|---|---|---|---|---|---|---|
| Variance inflation factor (VIF) Values | 2.601 | 2.515 | 2.117 | 2.720 | 1.438 | 1.644 | 1.675 | 3.078 |

**Note:** "PE: Performance Expectancy; EE: Effort Expectancy; SI: Social Influence; FC: Facilitating Conditions; HM: Hedonic Motivation; LC: Lifestyle Compatibility; PT: Perceived Trust; IWPD: Intention to Adopt WPDs.

**Source:** authors' data analysis."

## Method of data analysis

Partial least squares structural equation modelling (PLS-SEM) and artificial neural networks (ANN) were employed in the study as a part of the two-staged hybrid analytical approach. The PLS-SEM was used in the first stage to assess the model reliability, validity, and significant associations among the predictors of WPD adoption intention. In the second stage, ANN was applied to determine the significant importance of the factors of the adoption intention of WPD. The two approaches were compared to determine whether any variances were present in the factors used to predict WPD adoption intention. According to Chan and Chong [47], despite the frequent use of PLS-SEM in social and behavioural science to validate the relationship between independent and dependent variables, it is rarely incorporated with other artificial intelligence algorithms. Moreover, SEM could effectively manage complicated models, small samples, and non-normal data [48, 49]. Given that SEM is confined to the evaluation of linear models, the complexity may be oversimplified, making it unpredictable. The ANN method was used to establish non-linear relationships between the predictors in this study to mitigate this drawback. According to Chan and Chong [47], ANN improves understanding of complex non-linear and linear associations among components that affect technology adoption intentions. Compared to the traditional regression methods, ANN improves the accuracy of predictions [50]. The ANN analysis was also conducted for multi-layer perception to create normalised importance.

## Data analysis

**Demographic characteristics.**   Demographic details, including respondents' age, gender, average monthly income, marital status, education and resident city, are presented in Table 3.

## Reliability and validity

The construct reliability and validity were evaluated, as shown in Table 4. The minimum value of Cronbach's alpha value amounted to 0.860, which was higher than the minimum threshold value of 0.7 as per the suggestion by Hair et al. [51]. Overall, the factors were proven reliable. Based on the assessment of internal consistency using Composite Reliability (CR) with the values ranging from 0.898 to 0.966, all values were found to be higher than the minimum threshold value of 0.7 as per Hair et al.'s [51] recommendation. Therefore, the CR test indicated that all of the variables in this study dataset were highly reliable. Fornell and Larcker [52] suggested that convergent validity should be assessed through AVE using a threshold value higher than 0.50. The AVE values for all of the components in this study dataset ranged from 0.639 to 0.850, which was higher than the minimum suggested threshold. This result proved that convergence validity had a positive effect that supported the uni-dimensionality for each factor.

The Fornell-Larcker criterion, (Table 5) and cross-loading (Table 6) were employed in this study to provide a more thorough understanding of discriminant validity [53]. According to the Fornell-Larcker criterion, the square root of AVE for each construct should be higher than its strongest correlation with other constructs, which maintains the discriminant validity of the measurement model [48]. The Fornell-Larcker Criterion values of all of the components in this research dataset fulfilled the discriminant validity criteria. As a result, no lack of discriminant validity was found. Cross-loading is a technique for comparing the outer loadings of constructs, while previous research demonstrated that all factor loadings must be higher than 0.60 [51, 54]. An indicator's loading with its associated latent construct should surpass the loadings of all the remaining latent variables in cross-loadings [55]. All the factor loadings in the study dataset are more than 0.5, with all positive values exceeding the threshold value. Additionally,

**Table 3. Respondents demographic details.**

|  | N | % |  | N | % |
|---|---|---|---|---|---|
| *Gender* |  |  | *Average Monthly Income* |  |  |
| Male | 119 | 39.9 | Below RMB2500 | 96 | 32.2 |
| Female | 179 | 60.1 | RMB2501-5000 | 71 | 23.8 |
| Total | 298 | 100.00 | RMB5001-7500 | 56 | 18.8 |
|  |  |  | RMB7501-10000 | 32 | 10.7 |
| *Age* |  |  | More than 10000 | 43 | 14.4 |
| 20–25 Years | 150 | 50.3 | Total | 298 | 100.00 |
| 26–30 Years | 65 | 21.8 |  |  |  |
| 31–35 Years | 30 | 10.1 | *Educational Background* |  |  |
| 36–40 Years | 53 | 17.8 | Secondary school certificate | 37 | 12.4 |
| Total | 298 | 100.00 | Diploma certificate | 37 | 12.4 |
|  |  |  | Bachelor degree or equivalent | 143 | 48.0 |
| *Marital status* |  |  | Master's degree | 76 | 25.5 |
| Single | 186 | 62.4 | Doctoral degree | 5 | 1.7 |
| Married | 110 | 36.9 | Total | 298 | 100.00 |
| Divorced | 2 | .7 |  |  |  |
| Total | 298 | 100.00 | *Resident City* |  |  |
|  |  |  | Tier 1 Cities (Beijing, etc.) | 63 | 21.1 |
|  |  |  | Tier 2 Cities (Xi'an, etc.) | 116 | 38.9 |
|  |  |  | Tier 3 Cities (Jilin, Zhuhai, etc.) | 50 | 16.8 |
|  |  |  | Tier 4 Cities and rural areas | 69 | 23.2 |
|  |  |  | Total | 298 | 100.00 |

**Note:** RMB: Renminbi: official currency of the People's Republic of China.

**Source:** authors' data analysis.

all the factor loadings surpassed the remaining latent variables' loading values, which proved the strong discriminant validity of the study dataset.

## Path analysis

The coefficient of determination ($r^2$) reflects the degree of explained variance or the percentage of the variance in outcome variables that could be explained through a linear model. The

**Table 4. Reliability and validity.**

| Factors | No of Items | Cronbach's Alpha | Dijkstra and Henseler's rho | Composite Reliability | Average Variance Extracted |
|---|---|---|---|---|---|
| PE | 6 | 0.910 | 0.912 | 0.930 | 0.690 |
| EE | 5 | 0.860 | 0.869 | 0.898 | 0.639 |
| SI | 5 | 0.956 | 0.957 | 0.966 | 0.850 |
| FC | 5 | 0.867 | 0.871 | 0.904 | 0.653 |
| HM | 5 | 0.903 | 0.911 | 0.928 | 0.720 |
| LC | 5 | 0.908 | 0.912 | 0.932 | 0.733 |
| PT | 5 | 0.917 | 0.927 | 0.938 | 0.750 |
| IWPD | 6 | 0.938 | 0.939 | 0.951 | 0.764 |

**Note:** "PE: Performance Expectancy; EE: Effort Expectancy; SI: Social Influence; FC: Facilitating Conditions; HM: Hedonic Motivation; LC: Lifestyle Compatibility; PT: Perceived Trust; IWPD: Intention to Adopt WPDs.

**Source:** authors' data analysis."

**Table 5. Discriminant validity (Fornell-Larcker Criterion).**

|  | PE | EE | SI | FC | HM | LC | PT | IWPD |
|---|---|---|---|---|---|---|---|---|
| PE | 0.830 |  |  |  |  |  |  |  |
| EE | 0.723 | 0.799 |  |  |  |  |  |  |
| SI | 0.486 | 0.461 | 0.922 |  |  |  |  |  |
| FC | 0.649 | 0.655 | 0.606 | 0.808 |  |  |  |  |
| HM | 0.390 | 0.363 | 0.345 | 0.402 | 0.849 |  |  |  |
| LC | 0.400 | 0.398 | 0.497 | 0.390 | -0.028 | 0.856 |  |  |
| PT | 0.409 | 0.498 | 0.466 | 0.531 | 0.177 | 0.343 | 0.866 |  |
| IWPD | 0.646 | 0.579 | 0.656 | 0.708 | 0.376 | 0.500 | 0.599 | 0.874 |

**Note:** "PE: Performance Expectancy; EE: Effort Expectancy; SI: Social Influence; FC: Facilitating Conditions; HM: Hedonic Motivation; LC: Lifestyle Compatibility; PT: Perceived Trust; IWPD: Intention to Adopt WPDs.

**Source:** authors' data analysis."

range of 0.19 to 0.32 is considered weak explanatory power, while 0.33 to 0.66 is considered moderate explanatory power, and 0.67 and above denotes strong explanatory power of $r^2$ [55]. The $r^2$ value (0.680) indicates that all the predictors could explain a significant portion of the variation in IWPD (e.g., 68%), demonstrating a strong explanatory power. The $f^2$ measures the effect size, while the ranges of $f^2 \geq 0.02$, $f^2 \geq 0.15$, and $f^2 \geq 0.35$ represent small, medium, and large effect sizes, respectively [56]. Based on the analysis of the study dataset, $f^2$ values of PE, SI, FC, HM, and LC indicated a small effect size of the constructs over IWPD. In contrast, the $f^2$ value of PT denoted a relatively higher effect size than other variables.

The path coefficients and p-values for all associations are presented in Table 7. Based on the Table 7, the effect of PE, SI, FC, HM, LC, and PT on IWPD amounted to 0.234, 0.195, 0.250, 0.101, 0,157 and 0.239, respectively. The findings highlighted the positive effect of PE, SI, FC, HM, LC, and PT on IWPD, which was also statistically significant at a 5% level of significance. The EE demonstrated an unprecedented negative effect on IWPD, although it was not statically significant.

## Artificial neural network analysis

An artificial Neural Network is a robust and adaptable method. In contrast to other linear methods, this method does not require multivariate assumptions to be fulfilled (e.g., normality, homoscedasticity, linearity, and multicollinearity). Thus, ANN models are considered more precise and extensive compared to all other linear models [57]. At this phase of ANN, the root mean square of error (RMSE) values for both the training and testing datasets were used to determine the relative accuracy of the prediction (see Table 8). The small and close values in the test and training dataset indicate the study model's high precision and robust predictive power [15]. In this study, the values of RMSE ranged from 0.321 to 0.440 in the training dataset. The values ranging from 0.340 to 0.461 in the testing dataset were found to have small and close values, which established the strong predictive power and relative prediction accuracy.

Typical model sensitivity evaluations include 'one-at-a-time' simulations that individually evaluate the impact of each independent variable while overlooking the interactions with other independent variables [58]. In ANN, sensitivity analysis enables the evaluation of input variables in terms of the significance of their impact on the output variable and the identification of factors that could be omitted without compromising network quality and the critical key factors [59]. The mean importance of ANN sensitivity analysis (Table 9) in this study dataset

**Table 6. Loadings and cross-loading.**

| Code | PE | EE | SI | FC | HM | LC | PT | IWPD |
|---|---|---|---|---|---|---|---|---|
| PE1 | 0.798 | 0.555 | 0.356 | 0.554 | 0.298 | 0.305 | 0.300 | 0.499 |
| PE2 | 0.845 | 0.618 | 0.339 | 0.558 | 0.299 | 0.325 | 0.326 | 0.524 |
| PE3 | 0.804 | 0.587 | 0.298 | 0.527 | 0.293 | 0.311 | 0.336 | 0.517 |
| PE4 | 0.854 | 0.571 | 0.445 | 0.501 | 0.352 | 0.335 | 0.359 | 0.584 |
| PE5 | 0.842 | 0.615 | 0.528 | 0.539 | 0.337 | 0.366 | 0.381 | 0.560 |
| PE6 | 0.838 | 0.658 | 0.440 | 0.558 | 0.358 | 0.350 | 0.328 | 0.529 |
| EE1 | 0.524 | 0.827 | 0.255 | 0.499 | 0.254 | 0.296 | 0.373 | 0.430 |
| EE2 | 0.759 | 0.828 | 0.387 | 0.560 | 0.288 | 0.385 | 0.384 | 0.530 |
| EE3 | 0.524 | 0.827 | 0.235 | 0.518 | 0.222 | 0.261 | 0.347 | 0.386 |
| EE4 | 0.557 | 0.735 | 0.651 | 0.534 | 0.367 | 0.380 | 0.493 | 0.542 |
| EE5 | 0.453 | 0.775 | 0.187 | 0.482 | 0.288 | 0.208 | 0.353 | 0.357 |
| SI1 | 0.444 | 0.417 | 0.902 | 0.512 | 0.310 | 0.433 | 0.412 | 0.595 |
| SI2 | 0.464 | 0.436 | 0.914 | 0.583 | 0.349 | 0.436 | 0.411 | 0.581 |
| SI3 | 0.399 | 0.396 | 0.922 | 0.550 | 0.287 | 0.463 | 0.433 | 0.584 |
| SI4 | 0.457 | 0.449 | 0.939 | 0.568 | 0.306 | 0.496 | 0.452 | 0.625 |
| SI5 | 0.475 | 0.428 | 0.932 | 0.579 | 0.340 | 0.463 | 0.436 | 0.635 |
| FC1 | 0.522 | 0.570 | 0.487 | 0.783 | 0.305 | 0.369 | 0.391 | 0.503 |
| FC2 | 0.562 | 0.586 | 0.449 | 0.802 | 0.353 | 0.330 | 0.394 | 0.546 |
| FC3 | 0.484 | 0.491 | 0.594 | 0.799 | 0.267 | 0.380 | 0.431 | 0.574 |
| FC4 | 0.505 | 0.464 | 0.425 | 0.814 | 0.338 | 0.216 | 0.462 | 0.605 |
| FC5 | 0.552 | 0.550 | 0.496 | 0.841 | 0.358 | 0.295 | 0.457 | 0.620 |
| HM1 | 0.322 | 0.300 | 0.203 | 0.403 | 0.817 | -0.095 | 0.056 | 0.262 |
| HM2 | 0.357 | 0.372 | 0.362 | 0.404 | 0.873 | 0.042 | 0.167 | 0.351 |
| HM3 | 0.294 | 0.245 | 0.322 | 0.241 | 0.818 | 0.058 | 0.186 | 0.317 |
| HM4 | 0.332 | 0.293 | 0.285 | 0.304 | 0.885 | -0.041 | 0.120 | 0.276 |
| HM5 | 0.344 | 0.319 | 0.273 | 0.353 | 0.849 | -0.093 | 0.195 | 0.362 |
| LC1 | 0.280 | 0.298 | 0.391 | 0.282 | 0.069 | 0.810 | 0.255 | 0.369 |
| LC2 | 0.345 | 0.359 | 0.486 | 0.347 | 0.019 | 0.875 | 0.364 | 0.433 |
| LC3 | 0.400 | 0.412 | 0.487 | 0.390 | 0.044 | 0.897 | 0.314 | 0.458 |
| LC4 | 0.356 | 0.307 | 0.405 | 0.257 | -0.091 | 0.877 | 0.228 | 0.407 |
| LC5 | 0.323 | 0.318 | 0.356 | 0.375 | -0.148 | 0.817 | 0.297 | 0.458 |
| PT1 | 0.344 | 0.415 | 0.354 | 0.435 | 0.162 | 0.274 | 0.888 | 0.471 |
| PT2 | 0.367 | 0.457 | 0.415 | 0.494 | 0.121 | 0.355 | 0.847 | 0.544 |
| PT3 | 0.402 | 0.467 | 0.404 | 0.534 | 0.148 | 0.296 | 0.881 | 0.565 |
| PT4 | 0.198 | 0.343 | 0.287 | 0.307 | 0.102 | 0.203 | 0.849 | 0.381 |
| PT5 | 0.405 | 0.446 | 0.508 | 0.475 | 0.217 | 0.325 | 0.864 | 0.580 |
| IWPD1 | 0.574 | 0.517 | 0.578 | 0.641 | 0.312 | 0.432 | 0.581 | 0.886 |
| IWPD2 | 0.564 | 0.484 | 0.623 | 0.572 | 0.314 | 0.459 | 0.532 | 0.843 |
| IWPD3 | 0.543 | 0.505 | 0.588 | 0.643 | 0.325 | 0.415 | 0.533 | 0.908 |
| IWPD4 | 0.529 | 0.506 | 0.509 | 0.614 | 0.359 | 0.413 | 0.452 | 0.868 |
| IWPD5 | 0.568 | 0.510 | 0.545 | 0.620 | 0.338 | 0.449 | 0.496 | 0.874 |
| IWPD6 | 0.606 | 0.514 | 0.592 | 0.623 | 0.324 | 0.451 | 0.540 | 0.865 |

**Note:** "PE: Performance Expectancy; EE: Effort Expectancy; SI: Social Influence; FC: Facilitating Conditions; HM: Hedonic Motivation; LC: Lifestyle Compatibility; PT: Perceived Trust; IWPD: Intention to Adopt WPDs.

**Source:** authors' data analysis."

**Table 7. Path coefficients.**

| Hypothesis | | β-value | CI-Min | CI-Max | *t-value* | *p-value* | $r^2$ | $f^2$ | Decision |
|---|---|---|---|---|---|---|---|---|---|
| $H_1$ | PE→IWPD | 0.234 | 0.104 | 0.346 | 3.268 | 0.001 | | 0.069 | Supported |
| $H_2$ | EE→IWPD | -0.062 | -0.167 | 0.031 | 1.037 | 0.150 | | 0.005 | Rejected |
| $H_3$ | SI→IWPD | 0.195 | 0.089 | 0.282 | 3.350 | 0.000 | | 0.060 | Supported |
| $H_4$ | FC→IWPD | 0.250 | 0.139 | 0.375 | 3.439 | 0.000 | 0.680 | 0.078 | Supported |
| $H_5$ | HM→IWPD | 0.101 | 0.028 | 0.184 | 2.118 | 0.017 | | 0.022 | Supported |
| $H_6$ | LC→IWPD | 0.157 | 0.077 | 0.246 | 3.148 | 0.001 | | 0.049 | Supported |
| $H_7$ | PT→IWPD | 0.239 | 0.139 | 0.329 | 4.172 | 0.000 | | 0.115 | Supported |

**Note:** PE: Performance Expectancy; EE: Effort Expectancy; SI: Social Influence; FC: Facilitating Conditions; HM: Hedonic Motivation; LC: Lifestyle Compatibility; PT: Perceived Trust; IWPD: Intention to Adopt WPDs.

**Source:** authors' data analysis."

indicated that PT was the most important variable, followed by LC and HM. Fig 2 represents the ANN model, and Fig 3 offers the predicted values by observation.

## Discussion

The current work aims to explore the critical factors that form the intention to use the WPDs. The factors taken from UTAUT2 with additional factors of lifestyle compatibility and perceived trust added to the current study research model. Secondly, we aim to evaluate the UTAUT2 factors explaining the intention to adopt the WPDs with the two additional factors of lifestyle compatibility and perception of trust. Based on the analysis, the study suggests that the PE for WPDs was found to have a positive and statistically significant impact on the users' intention to use WPD. The result was consistent with the majority of previous research works in the areas of m-payments [8, 15, 17, 23], e-wallet [18], m-wallet [22], m-banking [29], and wearable payment devices [34]. Contactless payment systems have been widely accepted in China due to the consumers' belief that contactless payments are perceived as having the performance expectancy to deliver the best services to the users. Furthermore, EE had a negative and statistically insignificant effect on the intention to use WPD, indicating that the findings were inconsistent with earlier research [21, 22, 28, 30]. However, because Chinese customers

**Table 8. RMSE values of artificial neural networks (N = 298).**

| Network | Sample size (Training) | Sample size (Testing) | RMSE (Training) | RMSE (Testing) |
|---|---|---|---|---|
| 1 | 202 | 96 | 0.334 | 0.340 |
| 2 | 208 | 90 | 0.353 | 0.391 |
| 3 | 197 | 101 | 0.352 | 0.377 |
| 4 | 205 | 93 | 0.440 | 0.368 |
| 5 | 223 | 75 | 0.376 | 0.461 |
| 6 | 202 | 96 | 0.347 | 0.434 |
| 7 | 228 | 70 | 0.374 | 0.430 |
| 8 | 226 | 72 | 0.406 | 0.407 |
| 9 | 196 | 102 | 0.321 | 0.369 |
| 10 | 217 | 81 | 0.432 | 0.433 |
| | | Mean | 0.373 | 0.401 |
| | | Standard Deviation | 0.014 | 0.041 |

**Source:** Author's data analysis.

**Table 9. Sensitivity analysis.**

| Network | PE | EE | SI | FC | HM | LC | PT |
|---|---|---|---|---|---|---|---|
| 1 | 0.157 | 0.062 | 0.070 | 0.164 | 0.108 | 0.147 | 0.292 |
| 2 | 0.171 | 0.029 | 0.084 | 0.115 | 0.137 | 0.132 | 0.332 |
| 3 | 0.128 | 0.080 | 0.143 | 0.147 | 0.143 | 0.113 | 0.245 |
| 4 | 0.127 | 0.002 | 0.078 | 0.231 | 0.165 | 0.173 | 0.224 |
| 5 | 0.117 | 0.088 | 0.103 | 0.183 | 0.152 | 0.135 | 0.221 |
| 6 | 0.149 | 0.067 | 0.190 | 0.159 | 0.120 | 0.127 | 0.188 |
| 7 | 0.098 | 0.064 | 0.202 | 0.076 | 0.101 | 0.095 | 0.364 |
| 8 | 0.195 | 0.043 | 0.158 | 0.095 | 0.109 | 0.166 | 0.234 |
| 9 | 0.124 | 0.096 | 0.121 | 0.129 | 0.133 | 0.121 | 0.276 |
| 10 | 0.194 | 0.117 | 0.184 | 0.215 | 0.049 | 0.063 | 0.177 |
| Mean Importance | 0.1460 | 0.0648 | 0.1333 | 0.1514 | 0.1217 | 0.1272 | 0.2553 |

**Note:** "PE: Performance Expectancy; EE: Effort Expectancy; SI: Social Influence; FC: Facilitating Conditions; HM: Hedonic Motivation; LC: Lifestyle Compatibility; PT: Perceived Trust.

**Source:** authors' data analysis."

are already accustomed to various sorts of contactless payment systems, they may overlook the additional effort required to use WPDs.

The study finding indicated that SI had a substantial positive and significant influence on the intention to use WPDs. This result was in line with prior research works on M-payment [15, 21, 23, 30, 33], e-wallet [18], and m-money [27]. It could be indicated that the opinions

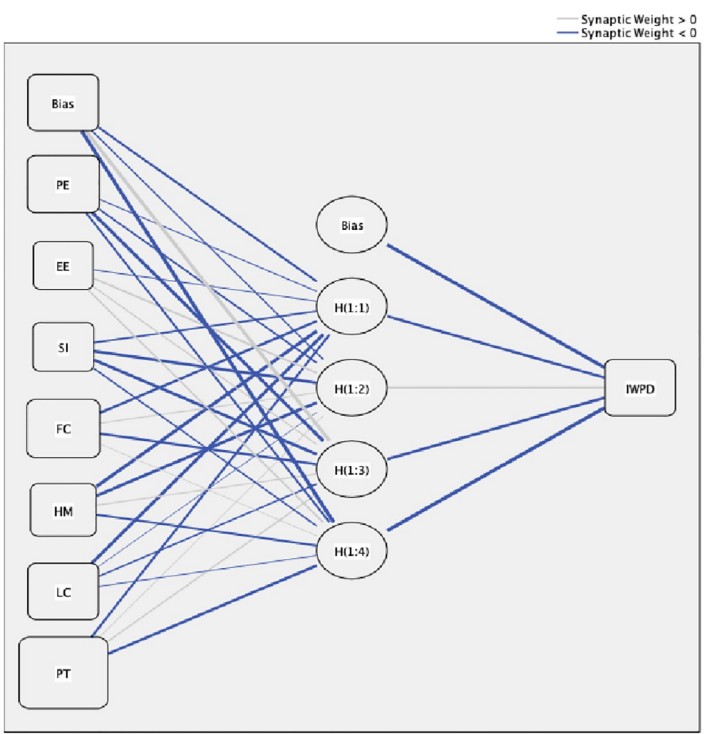

Hidden layer activation function: Hyperbolic tangent

Output layer activation function: Identity

**Fig 2. Hidden layer.**

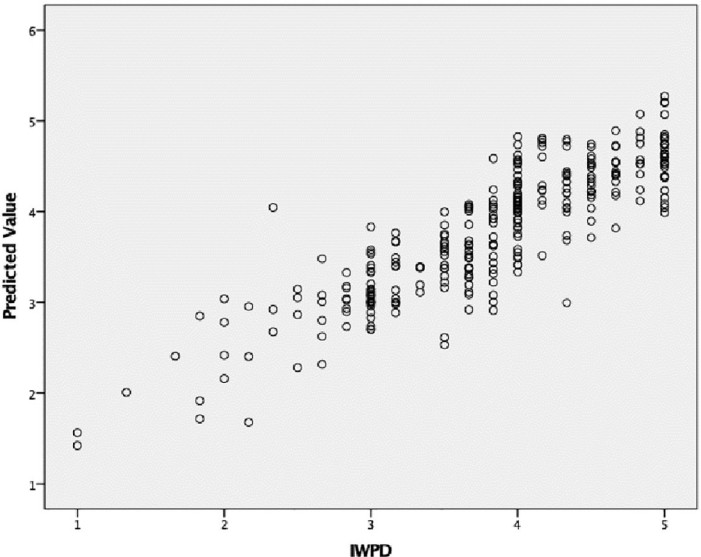

**Fig 3. Predicted by observed.**

and suggestions of prominent individuals in China strongly influence consumers' acceptance of WPD. This study also implied that Chinese people are more inclined to adopt WPD due to their respect for the recommendations from others parties (e.g., friends, peers, and coworkers).

The influence of FC on users' intention to adopt WPD was positive and statistically significant in this study. This result was in line with prior studies on contactless payments, including m-payment [20, 21, 23, 24], m-banking [29], and M-wallet [22], which indicated that FC had a significant impact on consumers' intention to accept contactless payments. The findings demonstrated that the potential to use WPD was necessarily contingent on the availability of facilities and support systems, although it could have more requirements. The significant impact of FC on users' intention to accept WPD was attributed to the fact that the facilities (e.g., smartphones, Internet access, 4G services, secured Wi-Fi, secure applications) and specific skill support (e.g., knowledge and guidelines to operating) required to use WPD are readily and frequently available to users.

In this study, HM was found to be a substantial predictor of intention towards WPD. Even though the result was in contrast to the majority of previous M-payment research works [15, 23, 24, 28], it was in line with other studies of m-payment [20, 30] and m-banking [29]. It was indicated from the results that Chinese consumers preferred the adoption of WPD and were pleased with the smooth, error-free, and simple transactions of WPD. In contrast to the consumers in other nations, Chinese WPD users found that the use of the new payment technology was enjoyable, exciting, and satisfying.

The impact of LC on the intention to use WPD has been a major topic in technology adoption studies. Findings indicated that LC had a substantial impact on the users' intentions for WPD adoption. This result was in line with the previous studies on e-wallets and m-wallet [18, 22] in the context of M-payment [12, 23, 28]. A higher LC would establish a better connection between the user's values, experiences, personality, and preferences with technology, resulting in a favourable attitude to accept it [60]. In addition, the WPD services were found to be well-received by users who perceived that the technology is suitable for all areas of their lifestyle and current living standards.

Users' behavioural intention to use WPD was recorded to be significantly influenced by PT. Provided that the WPD development was in the preliminary stage in China, a high level of ambiguity was found among the respondents regarding the payment method. Similar findings were recorded in the previous studies of m-payment [1, 15, 33], e-wallet, and m-wallet [18, 22], specifically the WPD study [34]. Customers opt to utilise the WPD that they regard with high confidence and consider authentic and reliable. This perception leads to improved assessments of products that deliver their expectations and are safe to use [18]. Overall, customers' inclination to use WPD is strongly influenced by their trust in it.

## Implication

### Theoretical implications

The main objective of the current study is examining factor affecting WPD adoption behaviour in China. In doing so, this study developed a more sophisticated framework based on a well-known UTAUT2) of Venkatesh et al. [26] to capture contributing factors to the formation of the intention to adopt WPD. Our model capture 68% of IWPD which confirms the validity of the UTAUT2 in context of WPD, and the findings highlighted UTUAT2 factors are important in formation of IWPD. Thus, it extends the literature of contactless payment. Furthermore, as mentioned earlier, wearable payment devices are currently in the early stages of commercialization and there is lack of studies in this context, our study contributes to the model and theoretical development in WPD adoption behaviour. Moreover, unlike previous studies on contactless payment adoption, this study included perceived trust and lifestyle compatibility as influencing factors of IWPD, which were never explored in the context of contactless payment especially wearable device. The significant influences of perceived trust and lifestyle compatibility adds new evidence to the literature in contactless payment.

In addition to SEM-PLS, the current study employed a two-step analytical approach (SEM-ANN) to confirm factors that contribute to the formulation of an IWPD. This ANN validated the excellent predictive accuracy of data fitness by focusing on the impacts of the most important variables. Such empirical estimation is still limited in previous research of contactless payments in China that used UTAUT2. The current study is the first attempt in using an integrated ANN model to observe the predictors of WPD adoption variables at the individual level. In our cases, the dual analytical strategy offered consistent findings and suggested that more efforts be undertaken to extend the UTAUT model to explore technology adoption. The ANN findings suggested that the four prime factors contributing to the formation of an intention to adopt WPDs are PT, FC and PE, respectively. Hence, it offers important contribution to the literature of contactless payment. Finally, the current study explored in the context of the China market thus improving literature of the China market and offering literature to compare with other countries.

### Practical implications

The primary goal of this study is to identify and validate a set of attributes that enhance the behavioural intention to use WPD systems in China. The current study offers a few practical implications for product innovators, financial institutions, and shoppers. The current study shows that PE, EE, FC, HM, LC, and PT have a substantial impact on WPD adoption. Such knowledge can assist financial institutions and device manufacturers in developing new plans and marketing methods to persuade WPD adopters. For example, relating to performance expectations, wearable payment device manufacturers and financial institutions should improve the functionalities and features of their devices, where they may focus on ensuring availability, performance, effectiveness, and efficiency, including 24*7 services and customer

support. Referring to effort efficiency, WPD should be easy-to-use with an adaptable user interface and a high-quality product experience. The significance of facilitating conditions suggests that WPD manufacturers should include a practical benefit and full instructions to be effectively conveyed to the consumers via marketing and social media to spread information and the advantages of WPD among users.

Furthermore, the significance of lifestyles implies that the adoption and usage of different models of WPD is likely to change in China with changing lifestyles and fashion trends because wearable device are currently deemed as fashion accessory. In market like China, owning a high-end technology device symbolize the high social status [61]. Hence, in an attempt to hold and increase market share in the WPD industry, device manufacturers should architect and develop new products with continuous market assessment where they need to assess how their products suit customers' lifestyles and current fashion trends. Additionally, device makers must evaluate the needs and lifestyles of customers in various segments in order to design and manufacture the products that best suit those customers. In advertising, WPD manufacturers and service providers should consider the living standards and lifestyle patterns of their target customer segments, where they may hire a fashion icon as their brand ambassador to promote products.

Moreover, trust has always been a critical key factor in financial services and technological products. The relevance of trust and security-related concerns is further highlighted by the discovery that perceived trust is a crucial determinant of the intention to use WPD. Hence, manufacturers and service providers should determine what customers perceive as trust-related risks associated with WPD usage in order to make WPD more trustworthy [15]. Therefore, in the first place, manufacturers should establish the safest possible infrastructure and environment for WPD services. Then, WPD providers should disclose information detailing data privacy rules, end-to-end encryption, biometric authentication capabilities (e.g., fingerprint readers), secured networks, and sensor-based procedures, which make customers feel trustworthy and protected. Additionally, they need to ensure that their WPD is error free in terms of software functioning. On the other hand, financial institutions, as a service provider, have to provide data security, service accessibility, platform security for wearable device payments, and user assurance that they can execute a variety of financial transactions efficiently and safely utilizing their wearable payment devices. Financial institutions may make use of a number of sophisticated encryption and authentication techniques to increase security. All these measures will ultimately increase trust of users towards device and servicer providers.

Finally, the findings would assist the government and key official policymakers in the governing bodies in improving the policies to enhance wide acceptance of WPD and accomplish the goal of developing a cashless society.

## Conclusion

The WPD technology is still in its inception stage, with a small number of research works being conducted to properly understand its potential in contactless payment. The WPD is gaining popularity as a viable alternative to cheques, cash, and debit/credit cards among customers and businesses. Despite the increasing popularity of wearable devices, the wearable payment technology sector remains in its infancy stage [62]. This study was performed to shed light on the intention of using WPD as a fast and flexible payment option. To fill the research gaps in the WPD context in China, this study used a comprehensive and integrated research model, which incorporated components from earlier studies such as perceived trust and lifestyle compatibility. Several similarities and differences were found between the current findings and previous contactless payment study findings, in which SI, LC, PT, and HM had a

substantial influence on WPD adoption intentions. Based on the ANN analysis, the essential factors influencing Chinese consumers' adoption of WPD were PT, FC, PE, LC, and HM, while EE were deemed unimportant. Additionally, there was no substitute found for a deeper understanding of the key constructs of adoption intention in terms of designing, refining, and implementing the WPD services, functionalities, and applications that could assist in the achievement of high customer satisfaction while adding value to the transition to a cashless society.

## Study limitations and future research options

However, three pertinent limitations were found in this study. Firstly, the use of cross-sectional design restricted the controllability of unobserved heterogeneity, preventing a strong foundation for proving causality. Therefore, future longitudinal research was suggested to allow variable arrangements to be designed, created, and measured more efficiently using data acquired over a long period of time. Secondly, while the current findings offered a better match in the use of two more variables (LC and PT), other important aspects such as innovativeness, risk, and brand image were overlooked despite their potential to provide a more thorough explanation of the adoption of WPD. By extending the UTAUT2 in the context of Chinese consumers' intention to use WPD, this study has offered prospects for academics and researchers to conduct future research that includes these components. Considering that UTAUT2 was used to examine behavioural intention, the study scope was broadened to include other behavioural intention theories (e.g., Diffusion of Innovation, Theory of Planned Behaviour) to study the WPD innovation adoption in China. Lastly, the study focused on young people in China who are financially stable and advanced technology users. Thus, future studies were suggested to examine the model focusing on various demographic populations from a cross-country perspective, which comprises diverse cultural, environmental, economic, and technological circumstances, to explore the behavioural intention of adopting WPD.

## Supporting information

**S1 Appendix.**
(DOCX)

**S1 Data.**
(SAV)

## Author Contributions

**Conceptualization:** Li Luyao, Abdullah Al Mamun, Naeem Hayat, Qing Yang, Mohammad Enamul Hoque, Noor Raihani Zainol.

**Data curation:** Li Luyao.

**Formal analysis:** Abdullah Al Mamun, Naeem Hayat, Noor Raihani Zainol.

**Methodology:** Li Luyao, Abdullah Al Mamun, Qing Yang, Mohammad Enamul Hoque.

**Writing – original draft:** Li Luyao, Naeem Hayat, Qing Yang.

**Writing – review & editing:** Abdullah Al Mamun, Mohammad Enamul Hoque, Noor Raihani Zainol.

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
