## [Decision Letter · Decision Letter 0]

27 Apr 2022

PONE-D-22-02148Predicting the Intention to Adopt Wearable Payment Devices in China: The Use of Hybrid SEM-Neural Network ApproachPLOS ONE

Dear Dr. Al Mamun,

Thank you for submitting your manuscript to PLOS ONE. After careful consideration, we feel that it has merit but does not fully meet PLOS ONE’s publication criteria as it currently stands. Therefore, we invite you to submit a revised version of the manuscript that addresses the points raised during the review process.

It is requested to (for details see the enclosed comments):

-present the model before the hypotheses development;

-rework on the correlation values of constructs in Table 3;

-expand the section "Theoretical implications";

- clarify the influence of PE on the intention;

- include in the section "Discussion" the answer to the three research questions stated in the Introduction;

-add a new section on limitations of the study and future developments;

- include a table presenting the items adopted for constructs' measurement;

- add the results of common methods variance;

- clarify in the section "Literature review" that the UTAUT2 constructs "price value" and "habit" were NOT considered in this study;

- include in the section "Discussion" some considerations about the possible effects of the sample composition on the results;

-revise the manuscript correcting typos (e.g. at the end of page 13 "Fndings").

We look forward to receiving your revised manuscript.

Kind regards,

Filomena Papa

Academic Editor

PLOS ONE

Journal Requirements:

4. Please ensure that you refer to Figure 1 and 2 in your text as, if accepted, production will need this reference to link the reader to the figure.

Additional Editor Comments:

UTAUT2 construct "price value" was not considered in this study. As a consequence the sentence "Following the aforementioned literature results and judgments, price value was taken into account in this study" (Section "Literature review", page 4) should be modified. Also the UTAUT2 construct "habit" was not included in the present study. On this respect the sentence "As a result, investigating the impact of habit was challenging, which led to the decision to adopt WPD" should be clarified.

Some considerations about the possible effects of the sample composition on the results should be included in the discussion. The results could be influenced by the composition of the sample e. g. the influence of FC on users’ intention to adopt WPD is not significant in this study with a sample composed by young people. Probably a different result could be obtained adopting a sample composed by people of a different age range.

Reviewers' comments:

Reviewer's Responses to Questions

**Comments to the Author**

1. Is the manuscript technically sound, and do the data support the conclusions?

Reviewer #1: Yes

Reviewer #2: Yes

2. Has the statistical analysis been performed appropriately and rigorously? 

Reviewer #1: Yes

Reviewer #2: Yes

3. Have the authors made all data underlying the findings in their manuscript fully available?

Reviewer #1: No

Reviewer #2: Yes

4. Is the manuscript presented in an intelligible fashion and written in standard English?

Reviewer #1: Yes

Reviewer #2: Yes

5. Review Comments to the Author

Reviewer #1: The paper is interesting and timely. However, it needs further improvement.

1. Please present the model before the hypotheses development.

2. You need to rework on the correlation values of constructs in Table 3. Some of them are very high. In fact, anything more than 0.8 is alarming and indicates that the two constructs are similar in nature. You have many of them more than .800. Please fix this issue.

3. The model says that PE in non-significant on intentions. Why? It's a matter of great concern as if wearable watches are not significant then why this all the hard work for?

4. Theoretical implications are weak. Please strengthen it further.

5. Please keep a section on limitations and future research separately.

6. You have mentioned that PE has a minor impact on intentions in discussion whereas it is found non-significant. This is not acceptable. Please provide the reasons for this non-significant impact. The rationale provided for it is not logical.

Reviewer #2: In the discussion section, it should answer the three research questions stated in the introduction section. For example, the mediating effects of behavioural intention on the relationship between the predictors and actual adoption.

Before indicating validity and descriptive analysis, I think that the authors need to uncover the measurement items. However, I cannot see any table that indicating the measurements.

Please add the results of common methods variance.

Why the author not applied PLSPredict in PLS-SEM?

6. PLOS authors have the option to publish the peer review history of their article (what does this mean?). If published, this will include your full peer review and any attached files.

Reviewer #1: No

Reviewer #2: No

---

## [Author Response · Author response to Decision Letter 0]

10 May 2022

Kindly refer to the Reply to reviewer(s) Comments document attached.

---

## [Decision Letter · Decision Letter 1]

2 Aug 2022

PONE-D-22-02148R1Predicting the Intention to Adopt Wearable Payment Devices in China: The Use of Hybrid SEM-Neural Network ApproachPLOS ONE

Dear Dr. Al Mamun,

Thank you for submitting your manuscript to PLOS ONE. After careful consideration, we feel that it has merit but does not fully meet PLOS ONE’s publication criteria as it currently stands. Therefore, we invite you to submit a revised version of the manuscript that addresses the points raised during the review process.

It is requested to revise the manuscript:

-addressing comments of Reviewer 1;

-correcting minor errors, for instance substituting in the abstract “(e.g., PT and LC)” with “(i.e. PT and LC)”.

We look forward to receiving your revised manuscript.

Kind regards,

Filomena Papa

Academic Editor

PLOS ONE

Reviewers' comments:

Reviewer's Responses to Questions

**Comments to the Author**

1. If the authors have adequately addressed your comments raised in a previous round of review and you feel that this manuscript is now acceptable for publication, you may indicate that here to bypass the “Comments to the Author” section, enter your conflict of interest statement in the “Confidential to Editor” section, and submit your "Accept" recommendation.

Reviewer #1: (No Response)

Reviewer #2: All comments have been addressed

Reviewer #3: All comments have been addressed

2. Is the manuscript technically sound, and do the data support the conclusions?

Reviewer #1: No

Reviewer #2: Yes

Reviewer #3: Yes

3. Has the statistical analysis been performed appropriately and rigorously? 

Reviewer #1: Yes

Reviewer #2: Yes

Reviewer #3: Yes

4. Have the authors made all data underlying the findings in their manuscript fully available?

Reviewer #1: No

Reviewer #2: Yes

Reviewer #3: Yes

5. Is the manuscript presented in an intelligible fashion and written in standard English?

Reviewer #1: Yes

Reviewer #2: Yes

Reviewer #3: Yes

6. Review Comments to the Author

Reviewer #1: This study aims to investigate the effects of core constructs of UTAUT along with lifestyle compatibility and perceived trust to understand consumer intention to adopt WPD. The data were collected from 298 consumers in China. The PLS-SEM and ANN analysis were conducted to analyse the data. The authors have come up with some interesting findings. However, the paper has a number of issues as well:

1. Inadequate research analysing the behavioral elements influencing adoption of WPD is not a compelling reason for undertaking this research. The research gap should be found out based on the gaps from the existing literature.

2. There is no need of bringing the selection of what constructs to be used in the proposed model under introduction section.

3. The data is very skewed and hence does not provide the full understanding of broader range of customers of various age groups. It generalisation is very limited. There is no meaning of understanding the adoption of any such technology for young generation as they are quite well-versed with these technologies anyway. You need to provide convincing rationale for undertaking this research. There is no novelty of doing it as everyone knows that such new technologies are well adopted within the younger generation anyway.

4. There is nothing new being reported based on the analysis. EE is coming non-significant as usual so are other variables as significant as well.

5. Discussion does not provide anything new from findings. Just saying that the findings are in line with the previous studies clearly saying that there is nothing new in the paper.

6. This is the reason why theoretical and practical implications are quite limited.

Reviewer #2: (No Response)

Reviewer #3: The statistical analysis in the current version of the paper is well done and described. I have no comments for the authors and I suggest accepting the paper in its current version.

7. PLOS authors have the option to publish the peer review history of their article (what does this mean?). If published, this will include your full peer review and any attached files.

Reviewer #1: No

Reviewer #2: No

Reviewer #3: No

---

## [Author Response · Author response to Decision Letter 1]

11 Aug 2022

Reply to Reviewer(s) Comments

PONE-D-22-02148R2

Comments to the Author

1. If the authors have adequately addressed your comments raised in a previous round of review and you feel that this manuscript is now acceptable for publication, you may indicate that here to bypass the “Comments to the Author” section, enter your conflict of interest statement in the “Confidential to Editor” section, and submit your "Accept" recommendation.

Reviewer #1: (No Response)

Reviewer #2: All comments have been addressed

Reviewer #3: All comments have been addressed

2. Is the manuscript technically sound, and do the data support the conclusions?

Reviewer #1: No

Reviewer #2: Yes

Reviewer #3: Yes

3. Has the statistical analysis been performed appropriately and rigorously? 

Reviewer #1: Yes

Reviewer #2: Yes

Reviewer #3: Yes

4. Have the authors made all data underlying the findings in their manuscript fully available?

Reviewer #1: No

Authors Reply: Kindly refer to the data provided as Supporting Information: Data - Predicting the Intention to Adopt WPDs.sav

Reviewer #2: Yes

Reviewer #3: Yes

5. Is the manuscript presented in an intelligible fashion and written in standard English?

Reviewer #1: Yes

Reviewer #2: Yes

Reviewer #3: Yes

6. Review Comments to the Author

Reviewer #1: This study aims to investigate the effects of core constructs of UTAUT along with lifestyle compatibility and perceived trust to understand consumer intention to adopt WPD. The data were collected from 298 consumers in China. The PLS-SEM and ANN analysis were conducted to analyse the data. The authors have come up with some interesting findings. However, the paper has a number of issues as well:

1. Inadequate research analysing the behavioral elements influencing adoption of WPD is not a compelling reason for undertaking this research. The research gap should be found out based on the gaps from the existing literature.

Authors Reply: Thank you for your suggestions prof. We amended our the motivation and research gap.

Amended Manuscript (Added):

Despite the usefulness and benefits of adopting wearable payment devices, there is a lack of study on wearable payment device adoption behaviour considering they are considerably in the early stages of commercialization. Interestingly, research in the context of China on m-payment usage (Cao et al., 2018; Chen et al., 2019; Huang et al., 2020) and wearable technology adoption (Chuah et al., 2016) has sought to uncover the factors driving technological product and service adoption. Furthermore, while m-payment and WPD serve the same purposes, there are significant technological and procedural differences that distinguish WPD from m-payment (Lee et al., 2020). Additionally, an individual’s behavioural factors toward different products and services are totally subjective, and thus their adoption behaviour is likely to be different for each. So, understanding of m-payment of adoption behaviour, it may not be useful for product innovators and shoppers of WPD. Identifying research gaps in wearable payment device adoption behaviour, the current study investigates factors impacting WPD adoption in China.

2. There is no need of bringing the selection of what constructs to be used in the proposed model under introduction section.

Authors Reply: Thank you for your suggestions prof. We removed the irrelevant part as recommended (discussion about constructs)

3. The data is very skewed and hence does not provide the full understanding of broader range of customers of various age groups. It generalisation is very limited. There is no meaning of understanding the adoption of any such technology for young generation as they are quite well-versed with these technologies anyway. You need to provide convincing rationale for undertaking this research. There is no novelty of doing it as everyone knows that such new technologies are well adopted within the younger generation anyway.

Authors Reply: Thank you for the comment. I understand your frustration, research in social science often feels like ‘mentioning the obvious’. If we criticise like this, we can criticise almost all the papers published in all journals. 

Saying that, I am not sure how to reply to this comment or what to improve based on this comment as there is nothing specific. 

4. There is nothing new being reported based on the analysis. EE is coming non-significant as usual so are other variables as significant as well.

Authors Reply: Again, there is nothing specific that we can do based on the comment above. 

5. Discussion does not provide anything new from findings. Just saying that the findings are in line with the previous studies clearly saying that there is nothing new in the paper.

Authors Reply: We have highlighted PT and LC as new findings of the our study in the context WPD. 

Amended Manuscript (Highlighted BLUE):

Typical model sensitivity evaluations include ‘one-at-a-time’ simulations that individually evaluate the impact of each independent variable while overlooking the interactions with other independent variables (Beres & Hawkins, 2001). In ANN, sensitivity analysis enables the evaluation of input variables in terms of the significance of their impact on the output variable and the identification of factors that could be omitted without compromising network quality and the critical key factors (Mrzygłód, 2020). The mean importance of ANN sensitivity analysis (Table 9) in this study dataset indicated that PT was the most important variable, followed by LC and HM. Figure 2 represents the ANN model, and Figure 3 offers the predicted values by observation. 

6. This is the reason why theoretical and practical implications are quite limited.

Authors Reply: Thank you. We have extended implications. 

Amended Manuscript (Highlighted BLUE):

Implication

Theoretical Implications

The main objective of the current study is examining factor affecting WPD adoption behaviour in China. In doing so, this study developed a more sophisticated framework based on a well-known UTAUT2) of Venkatesh et al. (2012) to capture contributing factors to the formation of the intention to adopt WPD. Our model capture 68% of IWPD which confirms the validity of the UTAUT2 in context of WPD, and the findings highlighted UTUAT2 factors are important in formation of IWPD. Thus, it extends the literature of contactless payment. Furthermore, as mentioned earlier, wearable payment devices are currently in the early stages of commercialization and there is lack of studies in this context, our study contributes to the model and theoretical development in WPD adoption behaviour. Moreover, unlike previous studies on contactless payment adoption, this study included perceived trust and lifestyle compatibility as influencing factors of IWPD, which were never explored in the context of contactless payment especially wearable device. The significant influences of perceived trust and lifestyle compatibility adds new evidence to the literature in contactless payment. 

In addition to SEM-PLS, the current study employed a two-step analytical approach (SEM-ANN) to confirm factors that contribute to the formulation of an IWPD. This ANN validated the excellent predictive accuracy of data fitness by focusing on the impacts of the most important variables. Such empirical estimation is still limited in previous research of contactless payments in China that used UTAUT2. The current study is the first attempt in using an integrated ANN model to observe the predictors of WPD adoption variables at the individual level. In our cases, the dual analytical strategy offered consistent findings and suggested that more efforts be undertaken to extend the UTAUT model to explore technology adoption. The ANN findings suggested that the four prime factors contributing to the formation of an intention to adopt WPDs are PT, FC and PE, respectively. Hence, it offers important contribution to the literature of contactless payment. Finally, the current study explored in the context of the China market thus improving literature of the China market and offering literature to compare with other countries. 

Practical Implications

The primary goal of this study is to identify and validate a set of attributes that enhance the behavioural intention to use WPD systems in China. The current study offers a few practical implications for product innovators, financial institutions, and shoppers. The current study shows that PE, EE, FC, HM, LC, and PT have a substantial impact on WPD adoption. Such knowledge can assist financial institutions and device manufacturers in developing new plans and marketing methods to persuade WPD adopters. For example, relating to performance expectations, wearable payment device manufacturers and financial institutions should improve the functionalities and features of their devices, where they may focus on ensuring availability, performance, effectiveness, and efficiency, including 24*7 services and customer support. Referring to effort efficiency, WPD should be easy-to-use with an adaptable user interface and a high-quality product experience. The significance of facilitating conditions suggests that WPD manufacturers should include a practical benefit and full instructions to be effectively conveyed to the consumers via marketing and social media to spread information and the advantages of WPD among users.

Furthermore, the significance of lifestyles implies that the adoption and usage of different models of WPD is likely to change in China with changing lifestyles and fashion trends because wearable device are currently deemed as fashion accessory. In market like China, owning a high-end technology device symbolize the high social status (Gao et al. 2014). Hence, in an attempt to hold and increase market share in the WPD industry, device manufacturers should architect and develop new products with continuous market assessment where they need to assess how their products suit customers’ lifestyles and current fashion trends. Additionally, device makers must evaluate the needs and lifestyles of customers in various segments in order to design and manufacture the products that best suit those customers. In advertising, WPD manufacturers and service providers should consider the living standards and lifestyle patterns of their target customer segments, where they may hire a fashion icon as their brand ambassador to promote products.

Moreover, trust has always been a critical key factor in financial services and technological products. The relevance of trust and security-related concerns is further highlighted by the discovery that perceived trust is a crucial determinant of the intention to use WPD. Hence, manufacturers and service providers should determine what customers perceive as trust-related risks associated with WPD usage in order to make WPD more trustworthy (Liébana et al., 2021). Therefore, in the first place, manufacturers should establish the safest possible infrastructure and environment for WPD services. Then, WPD providers should disclose information detailing data privacy rules, end-to-end encryption, biometric authentication capabilities (e.g., fingerprint readers), secured networks, and sensor-based procedures, which make customers feel trustworthy and protected. Additionally, they need to ensure that their WPD is error free in terms of software functioning. On the other hand, financial institutions, as a service provider, have to provide data security, service accessibility, platform security for wearable device payments, and user assurance that they can execute a variety of financial transactions efficiently and safely utilizing their wearable payment devices. Financial institutions may make use of a number of sophisticated encryption and authentication techniques to increase security. All these measures will ultimately increase trust of users towards device and servicer providers. 

 Finally, the findings would assist the government and key official policymakers in the governing bodies in improving the policies to enhance wide acceptance of WPD and accomplish the goal of developing a cashless society.

Reviewer #2: (No Response)

Reviewer #3: The statistical analysis in the current version of the paper is well done and described. I have no comments for the authors and I suggest accepting the paper in its current version.

Authors Reply: Thank you Prof.

---

## [Decision Letter · Decision Letter 2]

17 Aug 2022

Predicting the Intention to Adopt Wearable Payment Devices in China: The Use of Hybrid SEM-Neural Network Approach

PONE-D-22-02148R2

Dear Dr. Al Mamun,

We’re pleased to inform you that your manuscript has been judged scientifically suitable for publication and will be formally accepted for publication once it meets all outstanding technical requirements.

Kind regards,

Filomena Papa

Academic Editor

PLOS ONE

Additional Editor Comments (optional):

Reviewers' comments:

Reviewer's Responses to Questions

**Comments to the Author**

1. If the authors have adequately addressed your comments raised in a previous round of review and you feel that this manuscript is now acceptable for publication, you may indicate that here to bypass the “Comments to the Author” section, enter your conflict of interest statement in the “Confidential to Editor” section, and submit your "Accept" recommendation.

Reviewer #1: All comments have been addressed

Reviewer #2: All comments have been addressed

Reviewer #3: All comments have been addressed

2. Is the manuscript technically sound, and do the data support the conclusions?

Reviewer #1: Yes

Reviewer #2: Yes

Reviewer #3: Yes

3. Has the statistical analysis been performed appropriately and rigorously? 

Reviewer #1: Yes

Reviewer #2: Yes

Reviewer #3: Yes

4. Have the authors made all data underlying the findings in their manuscript fully available?

Reviewer #1: Yes

Reviewer #2: Yes

Reviewer #3: Yes

5. Is the manuscript presented in an intelligible fashion and written in standard English?

Reviewer #1: Yes

Reviewer #2: Yes

Reviewer #3: Yes

6. Review Comments to the Author

Reviewer #1: All my concerns have been addressed. I am happy with all the changes made by the authors in response to my queries in the last round of review. Please accept the paper in its current form.

Reviewer #2: Interesting writing and I enjoy reviewing this manuscript. Hybrid SEM studies and analysis are very useful for adding insight into quantitative studies

Reviewer #3: (No Response)

7. PLOS authors have the option to publish the peer review history of their article (what does this mean?). If published, this will include your full peer review and any attached files.

Reviewer #1: No

Reviewer #2: No

Reviewer #3: No

---

## [Editor Report · Acceptance letter]

19 Aug 2022

PONE-D-22-02148R2 

Predicting the Intention to Adopt Wearable Payment Devices in China: The Use of Hybrid SEM-Neural Network Approach 

Dear Dr. Al Mamun:

I'm pleased to inform you that your manuscript has been deemed suitable for publication in PLOS ONE. Congratulations! Your manuscript is now with our production department. 

Kind regards, 

on behalf of

Dr. Filomena Papa 

Academic Editor

PLOS ONE